# SEE IT FROM MY PERSPECTIVE: HOW LANGUAGE AFFECTS CULTURAL BIAS IN IMAGE UNDERSTANDING

**Amith Ananthram**$^\diamond$**, Elias Stengel-Eskin**$^\dagger$**, Mohit Bansal**$^\dagger$**, Kathleen McKeown**$^\diamond$
$^\diamond$Columbia University     $^\dagger$UNC Chapel Hill
amith@cs.columbia.edu

## ABSTRACT

Vision-language models (VLMs) can respond to queries about images in many languages. However, beyond language, *culture* affects *how* we see things. For example, individuals from Western cultures focus more on the central figure in an image while individuals from East Asian cultures attend more to scene context (Nisbett et al., 2001). In this work, we characterize the Western bias of VLMs in image understanding and investigate the role that language plays in this disparity. We evaluate VLMs across subjective and objective visual tasks with culturally diverse images and annotations. We find that VLMs perform better on the Western split than on the East Asian split of each task. Through controlled experimentation, we trace one source of this bias in *image understanding* to the lack of diversity in *language model* construction. While inference in a language nearer to a culture can lead to reductions in bias, we show it is much more effective when that language was well-represented during text-only pre-training. Interestingly, this yields bias reductions even when prompting in English. Our work highlights the importance of richer representation of all languages in building equitable VLMs. We make both our code and our models available at https://github.com/amith-ananthram/see-it-from-my-perspective.

## 1 INTRODUCTION

In *Ways of Seeing*, the art critic John Berger writes "The way we see things is affected by *what we know* and *what we believe*." This knowledge and these beliefs are informed by culture (Goldstein, 1957). Research from the cognitive sciences shows that culture mediates aspects of visual perception like color grouping and attentional focus (Chiao & Harada, 2008). While many studies have found that language models exhibit a Western worldview in knowledge (e.g. entities) and beliefs (e.g. values) (Xu et al., 2024), how vision-language models (VLMs) see things remains understudied. Whose perspective do state-of-the-art VLMs model? And how does language affect this perspective?

Recent progress in image-conditioned NLP has been driven by fusing pre-trained vision encoders with large language models (LLMs) to build VLMs (Awais et al., 2023). By leveraging the knowledge in their constituent LLMs, VLMs generalize to many image understanding tasks in a zero-shot fashion (Tsimpoukelli et al., 2021). Moreover, because their LLMs are often multilingual (Blevins & Zettlemoyer, 2022), some VLMs can respond fluently to queries about images in many languages.

However, while VLMs can generate text in *different languages*, they should also reflect *different cultures*, accurately modeling their imagery and perspectives (see Figure 1). This is not guaranteed through translation alone. Prior work has shown that vision encoders suffer degraded performance on non-Western images in objective tasks like object identification (De Vries et al., 2019; Richards et al., 2023; Nwatu et al., 2023) due to the distribution of images seen during multimodal fusion (Shankar et al., 2017; Pouget et al., 2024). In contrast, our focus is the effect of *language* on bias in both *objective* and *subjective* image understanding. **In particular, how does using a culturally closer language (during both text-only pre-training and prompting) affect this bias?**

We begin by evaluating off-the-shelf VLMs in multiple languages on tasks with images and labels from representative Western and East Asian cultures: object identification (Rojas et al., 2022), question answering (Schwenk et al., 2022; Romero et al., 2024), and art emotion classification (Mohamed et al., 2022). Then, to isolate the effect of language on bias in these tasks, we train our own

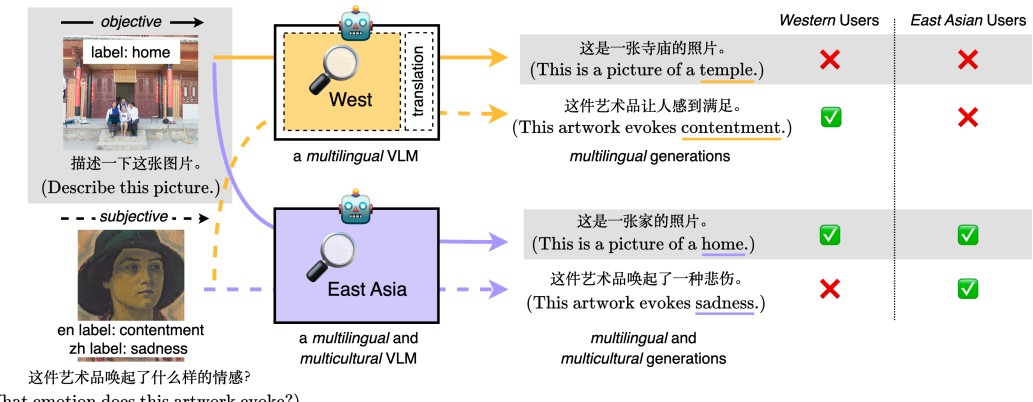

Figure 1: Whose perspective do VLMs model? Despite being *multilingual*, state-of-the-art VLMs exhibit a bias toward imagery and perspectives from Western culture. A more balanced language mix during text-only pre-training produces VLMs that are both *multilingual* and *multicultural*.

VLMs in a controlled setting. This restricts us to languages with publicly available LLMs that are comparable and strong enough to adapt to multimodal tasks: English and Chinese (Xu et al., 2024).

We find nearly all off-the-shelf VLMs exhibit a Western bias in every task, performing better on examples from both the narrow set of English-speaking countries and a broader set spanning North America / Western Europe (i.e. "the West", Spielvogel et al. (2012)) than on examples from China and its neighbors (i.e. East Asia, Choi (2010)). Moreover, this bias is not consistently reduced by prompting in Chinese instead of English even though Chinese is culturally closer to East Asia.

To understand this, we train comparable LLaVa variants (Liu et al., 2024) by fusing CLIP (Radford et al., 2021) with the 7B-parameter instruction-tuned variants of Llama2 and Baichuan2, which were pre-trained on 2 trillion (T) tokens of English and English/Chinese respectively (Touvron et al., 2023; Yang et al., 2023). We find that using a more representative language mix during LLM pre-training reduces model bias in visual tasks in both the narrow set of Chinese-speaking countries and the broader set of its East Asian neighbors, even when prompting in English. Moreover, while prompting in Chinese leads to reductions in bias, these reductions are much larger when Chinese is well represented during pre-training. These findings are more pronounced in subjective tasks like art emotion classification, but surprisingly, they hold for objective tasks such as visual question answering where we might otherwise expect the pre-training language mix to have no effect. Finally, while we might hope to see similar reductions by simply using a smaller-scale multilingual fusion corpus, we show this is no replacement for an LLM pre-trained on large quantities of Chinese text.

These results have important implications for the development of VLMs. Though we might hope that models leverage culturally-relevant associations when prompted in a language closer to a target culture, they do so much more effectively when that language was common during LLM pre-training. AI risks perpetuating hegemonies online, resulting in algorithmic monoculture (Bender et al., 2021). In fact, Western bias in VLMs is being exacerbated as models scale (Richards et al., 2023). Truly representative multimodal AI requires investment in multilingual foundation models.

**Note:** To measure **Western / East Asian bias**, we evaluate VLMs on examples with images and annotations from as many Western / East Asian countries as exist in published benchmarks (see Table 1). This includes both English / Chinese-speaking countries and their geographical and cultural neighbors (Spielvogel et al. (2012), Choi (2010)). While our controlled study of how language affects this bias is limited to only English and Chinese due to a lack of comparable multilingual LLMs, our evaluation examples span this more representative set of countries. Thus, we refer to our objects of study as *Western* and *East Asian* bias as a convenient shorthand.

Figure 2: Our approach. **Step 1**: We measure the Western bias of off-the-shelf ($OTS_i$) VLMs on *culturally diverse* image understanding tasks by comparing their performance on each task's Western and East Asian splits. **Step 2**: We train comparable multilingual VLMs (mLLaVA). We explore three model design choices focused on *language*: *(A)* the language mix in the pre-training corpus of the base LLM; *(B)* the prompting language; and *(C)* the language mix in the multimodal fusion corpus. We test each mLLaVA variant, measuring the effects of *(A)*, *(B)*, and *(C)* on Western bias.

In summary, our contributions are:

1. We show that VLMs exhibit a Western bias on both subjective and objective visual tasks.

2. We trace this Western bias to the language mix seen by VLMs during text-only pre-training; a more balanced mix reduces VLM bias, even when prompting in English.

3. While prompting in a culturally closer language can reduce Western bias in some VLMs, it is much more effective when that language is well represented during pre-training.

4. Both a balanced pre-training language mix and prompting in a culturally related language yield significant reductions in bias for both objective and subjective visual tasks.

5. A multilingual fusion corpus is not a substitute for pre-training with large amounts of multilingual text; addressing VLM bias requires intervention early in model development.

## 2 RELATED WORK

Cognitive science has shown that individuals from Western and East Asian cultures exhibit regular differences in perception when categorizing colors, grouping objects and attending to images (Bornstein, 1975; Goh et al., 2007; Nisbett et al., 2001).

In the language only setting, prior work has probed models for such cultural alignment, finding that they exhibit a Western bias in the world knowledge (Naous et al., 2023) and values (AlKhamissi et al., 2024) they encode. In the multimodal setting, the focus has been on characterizing Western bias in object detection and scene understanding (De Vries et al., 2019; Liu et al., 2021; Richards et al., 2023; Nwatu et al., 2023), tracing it to the distribution of images used for multimodal fusion (Shankar et al., 2017; Pouget et al., 2024). Proposed methods for reducing this bias rely on painstakingly constructing static knowledge bases (Yin et al., 2023) from Wikipedia (which can reflect this same Western lens, Naous et al. (2023)) or complicated prompting pipelines tightly coupled to a single task (Song et al., 2024). Moreover, many methods assume that diverse concepts from different cultures are interchangeable (for example the Chinese *erhu*, a string instrument, and Western drums), potentially stripping away culturally specific nuance (Zeng et al., 2023; Li & Zhang, 2023).

In contrast, we measure Western bias in both objective and subjective tasks where we want to *preserve* cultural differences. Moreover, while prior work has emphasized the effect of the pre-training image distribution on image understanding, we investigate the effects of the pre-training *language* distribution. Language reflects the lived experience of the people who use it (Andreas, 2022). Ye et al. (2023) find that human and machine captions exhibit language-specific regularities. We extend their work to other image understanding tasks, focusing on cultural regularities and the language modeling choices that produce them. As most text online is unimodal (Villalobos et al., 2022), this understanding allows us to better leverage this valuable resource to build more representative VLMs.

## 3 METHODS

We illustrate our approach in Figure 2. In step 1, *Bias Characterization*, we evaluate off-the-shelf (OTS) VLMs on *culturally diverse* visual tasks. We measure Western bias by comparing their performance on each task's Western and East Asian splits. In step 2, *Bias Sourcing*, we train comparable multilingual VLMs (mLLaVA), isolating the effects of language modeling choices on Western bias.

**Tasks** We select three tasks to evaluate Western bias in image understanding (Figure 3): the objective object identification and question answering and the subjective art emotion classification. Each task exhibits *image diversity* (from diverse places) or *label diversity* (from diverse annotators).

For each task $T$, we use images and annotations from North America and Western Europe as our Western split, $X_{\text{west}}, Y_{\text{west}}$; and we use images and annotations from East Asia as our East Asian split, $X_{\text{east}}, Y_{\text{east}}$. For each split, we include as many countries as are available across published benchmarks. Moreover, we assume that both the place an image was taken and the language spoken by an annotator are reasonable proxies for culture, a simplification made in other work as it is otherwise difficult to stratify evaluation tasks by culture (Goldstein, 1957; Ye et al., 2023).

In the non-Western setting, we focus on the culture of East Asia for several reasons. First, due to its inclusion of China, it is the highest resourced of all non-Western cultures. Thus, the bias we measure here should be a floor for the bias relative to other cultures; as other cultures are less resourced, the problem should be worse. Second, our goal is to identify the effect of *language* on any Western bias we measure. This requires controlled experimentation with LLMs pre-trained on equivalently large quantities of non-Western language. This restricts us to Chinese and East Asia (Xu et al., 2024).

**Defining Bias** Our focus is *error disparity* – given an evaluation metric $M$ and a task $T$ with two disjoint sets $X_A, Y_A$ and $X_B, Y_B$ (where $X$ are inputs, $Y$ are their labels and $A, B$ are values of attribute $\mathcal{R}$), a model $f_\theta$ that is unbiased with respect to $\mathcal{R}$ should perform similarly on $X_A$ and $X_B$ (Dwork et al., 2012; Shah et al., 2019; Czarnowska et al., 2021). That is, we expect:

$$M(f_\theta(X_A), Y_A) = M(f_\theta(X_B), Y_B) \tag{1}$$

Their ratio is the bias of $f_\theta$ with respect to $\mathcal{R}$. Here, $\mathcal{R}$ is culture, and our metric for Western bias is:

$$\text{bias}_{f_\theta} = \frac{M(f_\theta(X_{\text{West}}), Y_{\text{West}})}{M(f_\theta(X_{\text{East Asia}}), Y_{\text{East Asia}})} \tag{2}$$

which measures how many times as good $f_\theta$ is on the Western split of $T$ as on its East Asian split. For example, if $f_\theta$ were unbiased at task $T$, its accuracy should be similar on both splits so $\text{bias}_{f_\theta} = 1$. However, if its accuracy were twice as high on $T$'s Western split, $\text{bias}_{f_\theta} = 2$.

### 3.1 STEP 1: BIAS CHARACTERIZATION

We choose a representative set of $n$ off-the-shelf VLMs, $\{\text{OTS}_i : 1 \leq i \leq n\}$ and evaluate each one on our selected tasks. Then, we measure the Western bias of each $\text{OTS}_i$ ("Step 1" in Figure 2).

We evaluate each $\text{OTS}_i$ in a zero-shot setting as most datasets for these tasks skew Western (i.e., images from Western countries annotated by English speakers). Fine-tuning on such a dataset would complicate our evaluation of the underlying VLM's bias. Additionally, given the broader trend toward large general purpose VLMs we believe that the zero-shot setting is also the most realistic.

We evaluate each $\text{OTS}_i$ twice, with English and Chinese prompts. This allows us to test whether VLMs model the perspective of a culture more effectively when queried in a related language.

### 3.2 STEP 2: BIAS SOURCING

While we can characterize Western bias in existing VLMs, this does not allow us to make causal claims regarding its source. Each of the VLMs we evaluate differ from one another across many different axes: parameters, pre-training corpora and task-specific fine-tuning.

Thus, to disentangle the effects of language modeling choices, we train our own comparable VLMs (**mLLaVA**). We follow the training recipe of LLaVa: we fuse a pre-trained image encoder to a pre-trained base LLM by training on a multimodal fusion corpus of paired images and text (Liu et al.,

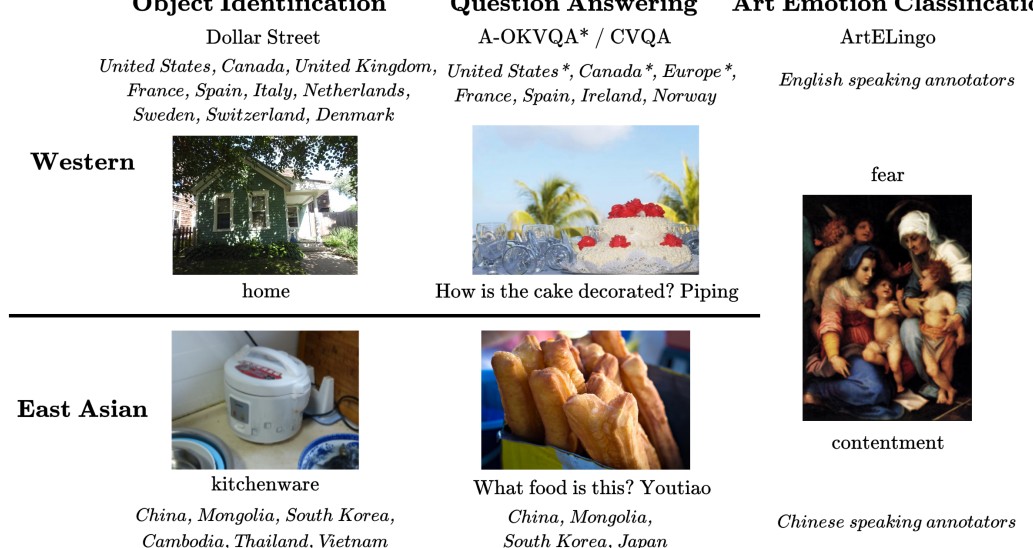

Figure 3: Examples from our *culturally diverse* image understanding tasks which range from the *objective* to the *subjective*: object identification, question answering and art emotion classification.

2024). To allow direct comparison, we keep everything fixed and systematically explore three axes of variation (the letters below correspond to **mLLaVA** elements in "Step 2" of Figure 2):

A) the language composition during pre-training of the base LLM (for the same # of tokens)

B) the prompting language during inference (for semantically equivalent prompts)

C) and, the language composition (monolingual/bilingual) during multimodal fusion (for the same images and semantic content).

**Measuring Reductions in Bias**    Models have different strengths – one might be better than another at both the Western and East Asian split of a particular task. This task-specific strength stems from the genres represented in their pre-training (Tiong et al., 2024) and is orthogonal to bias reduction. Model $A$ could be much better on the Western split of a task and only slightly better on the East Asian split when compared to another model $B$. Even though $A$ performs better than $B$ on the East Asian split due to its strength in that task, it is more biased than $B$. Therefore, when evaluating the effects of modeling choices, we measure the resultant *reduction in bias*.

## 4 EXPERIMENTS

### 4.1 TASKS

For each of our tasks, we select relevant portions of standard benchmarks to be our Western and East Asian splits (Table 1). For object identification and question answering, each split is broken into a narrow language-focused subset of only English and Chinese speaking countries ("en/zh") and an expansive regional subset ("region") of every available country from North America/Western Europe and East Asia. For examples, see Figure 3. Further details can be found in Section A.1.

**Object Identification: Dollar Street**    To measure Western bias in object identification, we use Dollar Street (Rojas et al., 2022). It contains images of 289 types of everyday objects manually labeled and uploaded by individuals from around the world (exhibiting *image diversity* and *label diversity*). As the labels include "toothbrush", "toilet" and "tv", we consider this task to be *objective*.

Chinese prompts and labels were hand-translated by native speakers. We evaluate VLMs in an open generation setting. As some images have multiple labels, we judge generations permissively using an LLM, granting credit for matching any correct label. Details can be found in Section A.1.1.

Table 1: Our tasks: object identification (Obj. Id.), visual question answering (VQA) and art emotion classification (Art). ID and LD specify image and label diversity (↓: low, ↑: high). "Prompts", the source of English and Chinese prompts (HW: hand-written, MT: machine translated). "Eval", how each task is evaluated (LM: LLM-as-a-Judge, MC: multiple choice). "Western" and "East Asian", the countries/languages in each task's splits. For examples from each task, see Figure 3.

| | ID / LD | Prompts en | Prompts zh | Eval | Subset | Western | East Asian |
|---|---|---|---|---|---|---|---|
| **Obj. Id.** | ↑ ↑ | HW | HW | LM | *en/zh* | US, Canada, UK | China |
| | | | | | *region* | + France, Spain, Italy, Switzerland, Denmark, Netherlands, Sweden | + Mongolia, S. Korea, Vietnam, Cambodia, Thailand |
| **VQA** | ↑ ↑ | HW | MT | MC | *en/zh* | A-OKVQA | China |
| | | | | | *region* | + Spain, France Ireland, Norway | + Mongolia, S. Korea, Japan |
| **Art** | ↓ ↑ | HW | HW | MC | *opp val all* | English speaking annotators | Chinese speaking annotators |

**Question Answering: AOKVQA & CVQA** To measure Western bias in question answering, we use A-OKVQA and CVQA (Schwenk et al., 2022; Romero et al., 2024). A-OKVQA contains images from MSCOCO queried from websites like Flickr for Western concepts from ImageNet (Lin et al., 2014). 80%+ of the images are from North America or Europe (Kirillov et al., 2023). Questions and 4 candidate answers (1 correct, 3 distractors) were written by English speakers on Mechanical Turk. CVQA is a complementary benchmark whose images and annotations are drawn from around the world. Annotators from 28 countries identified culturally relevant images and then annotated those images with questions and 4 candidate answers in both their native language and in English. Thus, the images, questions and answers in these benchmarks exhibit *image diversity* and *label diversity*. As these questions have globally correct answers, we consider VQA to be *objective*.

As our questions and answers are in English, to allow inference in Chinese, we translate questions and candidate answers with the Google Translate API. A native Chinese speaker evaluated 50 of these translations, assigning them an average score of 3.82 on a 4-point Likert scale, indicating high quality. We evaluate VLMs in a multiple-choice setting, using another LLM to select the closest candidate answer from each generation to calculate accuracy. Details can be found in Section A.1.2.

**Emotions in Artwork: ArtELingo** To measure Western bias in art emotion classification, we use the ArtELingo corpus (Achlioptas et al., 2021; Mohamed et al., 2022). ArtELingo contains emotion labels and textual rationales from English and Chinese speaking annotators for 80, 000 works from WikiArt. It covers 9 emotions: 4 positive (amusement, awe, contentment, excitement), 4 negative (anger, disgust, fear, sadness) and a catch-all, "something else". Each work of art receives 5 annotations for each language. While this task is *subjective*, we consider only the works of art with moderately high agreement in a language (≥ 3 annotators) to ensure cultural consensus. Additionally, we ignore all cases where the majority label was "something else". The artwork in ArtELingo was mostly created by Western artists so it exhibits low *image diversity* but high *label diversity*.

Of particular interest is art assigned a positive emotion by English annotators and a negative emotion by Chinese annotators (or vice versa). For this *opposite valence* subset, we know the disagreement extends beyond imperfect translation of emotion labels. Annotators perceived the art differently.

We evaluate VLMs on this task in an open-generation setting and select the candidate emotion closest to each generation using another LLM. Details can be found in Section A.1.3.

## 4.2 STEP 1: BIAS CHARACTERIZATION

**Off-the-Shelf Models: OTS$_i$** We select VLMs that differ along multiples axes (pre-training corpus, parameter count, etc.): the English BLIP2 (Flan-T5-XXL) (Li et al., 2023), LLaVA-1.5 (Llama-7B & Mistral-7B)[1] (Liu et al., 2023); the bilingual English/Chinese Qwen-VL (7B & 7B-Chat) (Bai et al., 2023); and the multilingual mBLIP (Bloomz-7B) (Geigle et al., 2023).

---

[1]https://huggingface.co/SkunkworksAI/BakLLaVA-1

**Evaluating Generations:** $M$   As we generate open-ended text from our off-the-shelf models $OTS_i$ and our **mLLaVA** variants for Object Identification, we evaluate this particular task using an LLM-as-a-Judge, specifically `Prometheus-2-8x7B` (Kim et al., 2024). Recent work has shown that LLM judgments are highly correlated with human ratings, especially for the short texts compared in image understanding tasks (Zheng et al., 2024; Ging et al., 2023). We use Prometheus-2 instead of GPT-4 as it is significantly cheaper and perfectly reproducible (the weights are fixed and published). We validate its effectiveness by comparing its judgments to those from native English and Chinese speakers for a sample of our generations. We find that our judge LLM agrees with our human raters between $88 - 89\%$ of the time in English and Chinese with Cohen's $\kappa$ values between $0.65 - 0.66$ indicating substantial agreement (Cohen, 1960). Additional details can be found in Section A.2.1.

For VQA and Art Emotion Classification, which are both multiple choice tasks, we use `Mistral-7B-Instruct-v0.2` to extract the candidate answer closest to the one described in each generation rather than using an LLM-as-a-Judge (Jiang et al., 2023). This extracted answer is used to calculate accuracy/F1. Additional details for these tasks are in Sections A.2.2 and A.2.3.

### 4.3   STEP 2: BIAS SOURCING

**Base LLMs**   To measure the effect of a base LLM's pre-training language mix on bias in image understanding, we choose two LLMs with different mixes but that are otherwise comparable: Llama2 and Baichuan2 (Touvron et al., 2023; Yang et al., 2023). We use their 7B parameter, instruction-tuned ("Chat") variants, each pre-trained on 2T tokens. Their most significant difference is the language mix during pre-training: Llama2 was trained on mostly English while Baichuan2 was trained on a balanced mix of English and Chinese. Additional details can be found in A.3.1.

**Fusion Corpus Construction**   For each of our base LLMs, we train English, Chinese and bilingual (English/Chinese) variants, requiring 3 comparable multimodal fusion corpora. For English, we use the fusion corpus constructed for the original LLaVA models: $595,375$ image-text pre-training examples from CC3M and $157,712$ synthetic visual instructions for images from MS-COCO (Lin et al., 2014; Liu et al., 2024). For Chinese, we use a publicly available machine translation[2] of this corpus which we ask a native speaker to evaluate. They assigned a random sample of $50$ translations an average score of $3.76$ on a $4$-point Likert scale, indicating high quality. For our bilingual corpus, we randomly sample half of each monolingual corpus, ensuring no image overlap. This results in 3 fusion corpora with semantically equivalent captions in English or Chinese for the same images.

**Training Details**   We train 6 **mLLaVA** variants (2 base LLMs $\times$ 3 fusion corpora) using the publicly released LLaVA recipe (Liu et al., 2024). Additional details can be found in Section A.3.

## 5   RESULTS

### 5.1   DO STATE-OF-THE-ART VLMS EXHIBIT A WESTERN BIAS IN IMAGE UNDERSTANDING?

In Figure 4, we plot the Western bias (Western performance divided by East Asian) of our off-the-shelf VLMs when prompted in English (●) and Chinese (✕) (full results are in the Appendix, Table 5). **90%+ of these VLMs exhibit a Western bias on each task** – they lie in the gold region. This holds for objective (object identification, VQA) and subjective tasks (art emotion classification).

Moreover, in **only 15/30 model task evaluations, Western bias is reduced when prompting in Chinese** (the Chinese ✕ appears to the left of the English ●). 3 models have reduced bias on Dollar Street, 3 have reduced bias on VQA, and 2 have reduced bias on ArtELingo. **While some VLMs are less biased when prompted in Chinese, this is not consistently true for all**.

Models like Qwen-VL that benefit from Chinese prompting in one task tend to see benefits in other tasks as well. This suggests that leveraging Chinese for better East Asian image understanding is a model-specific quality. It is also instructive to consider which models exhibit the least bias in English (i.e., with leftmost ●). mBLIP is one such example (Geigle et al., 2023). One defining characteristic of this VLM is that its base LLM is the multilingual Bloomz (Muennighoff et al.,

---

[2]https://huggingface.co/datasets/LinkSoul/Chinese-LLaVA-Vision-Instructions

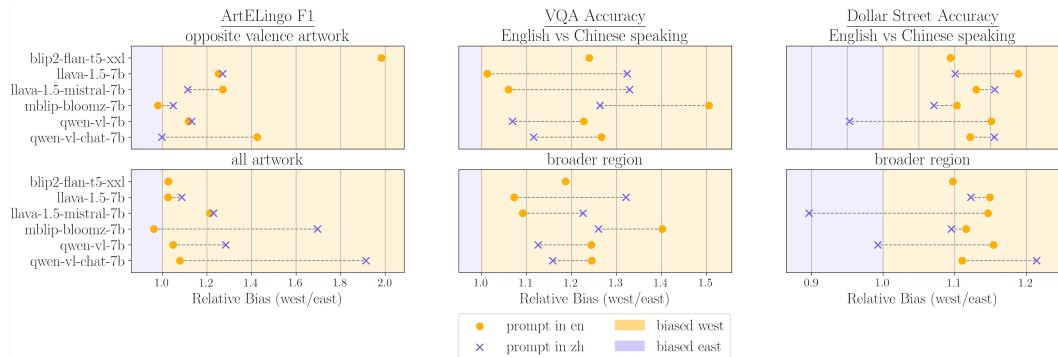

Figure 4: The bias (Western performance divided by East Asian performance) of each of our $OTS_i$ VLMs when prompted in English (●) and Chinese (✕). Markers that fall in the gold / purple regions indicate Western and East Asian biases respectively. **While Western bias reductions are seen across all tasks when prompting in Chinese, they are not seen consistently (in only 15/30 cases).**

2022). It is possible it uses more diverse associations from its multilingual pre-training corpus during multimodal inference, even in English. We investigate this further in the next section.

## 5.2 STEP 2: BIAS SOURCING

First, we note that base LLMs endow their VLMs with strengths in different tasks. When a Llama2-based model is better than the Baichuan2-based model in one split (West or East) for a particular task, it is generally better at the other split too. For example, in English, the Llama2-based models are better at object identification while Baichuan2-based models are better at art emotion classification.

However, our focus is bias as measured by the ratio between each model's scores on a task's Western and East Asian splits (Equation 2). A model with a bias of 1 is unbiased while 2 means it performs twice as well on the Western split. To study how bias changes with our three language modeling choices, we measure the difference in bias between two comparable mLLaVA variants, calculating significance using a two-factor design (Krzywinski, 2014). We plot these reductions along the $x$-axis in Figures 5a, 5b and 5c. Absolute scores (with $p$ values) are in the Appendix, Table 6.

### 5.2.1 HOW DOES THE LLM LANGUAGE MIX AFFECT BIAS IN IMAGE UNDERSTANDING?

We find that **pre-training the base LLM on a more balanced mix of languages (that is, more Chinese as for Baichuan2) reduces Western bias in image understanding** (Figure 5a).

We observe **significant reductions in bias for both *objective* and *subjective* tasks when prompting in Chinese**. For VQA, we see significant reductions in Western bias from 1.50 to 1.17 (Western accuracy went from 1.50 times the East Asian accuracy to 1.17 times) on the fine-grained English/Chinese-speaking country subset and reductions of 1.29 → 1.16 on the coarse-grained regional subset. On the subjective ArtELingo, we see significant reductions in Western bias of 2.47 → 1.23 on the subset where English and Chinese speakers disagree on the valence of the emotion ("opposite valence") and 1.82 → 1.16 on the full benchmark ("indicated by all").

We also see that more Chinese during text-only pre-training significantly **reduces bias for subjective tasks even when prompting in English**. On ArtELingo, we see reductions of 3.45 → 2.71 and 2.20 → 1.98 on the "opposite valence" and "all" subsets respectively.

### 5.2.2 HOW DOES THE PROMPTING LANGUAGE AFFECT BIAS IN IMAGE UNDERSTANDING?

**Prompting in a culturally closer language reduces Western bias on objective & subjective visual tasks, especially if the language was common during LLM pre-training** (Figure 5b).

With the mostly English Llama2 as our base LLM, we see minimal reductions in Western bias on VQA when prompting in Chinese. In contrast, using the bilingual Baichuan2 as our base LLM, we see a significant reduction in bias for VQA of 1.45 → 1.17 on the "en/zh" subset and a reduction

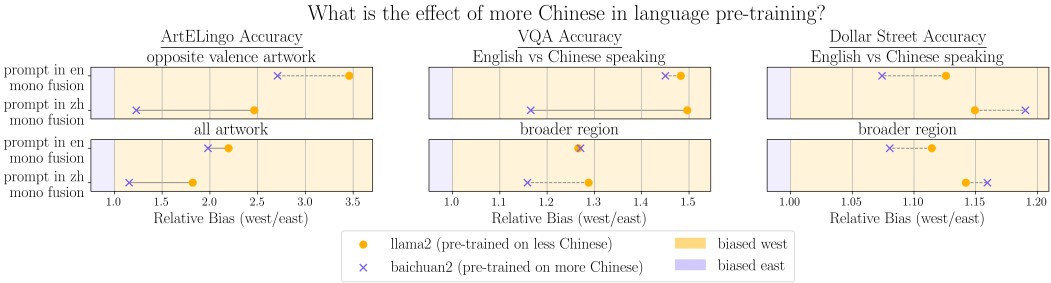

(a) ● indicates that the base LLM is the mostly English Llama2, ✕ the bilingual (English/Chinese) Baichuan2. **More Chinese during text-only pre-training reduces Western bias on objective and subjective tasks when prompting in Chinese; it also reduces Western bias on subjective tasks when prompting in English.**

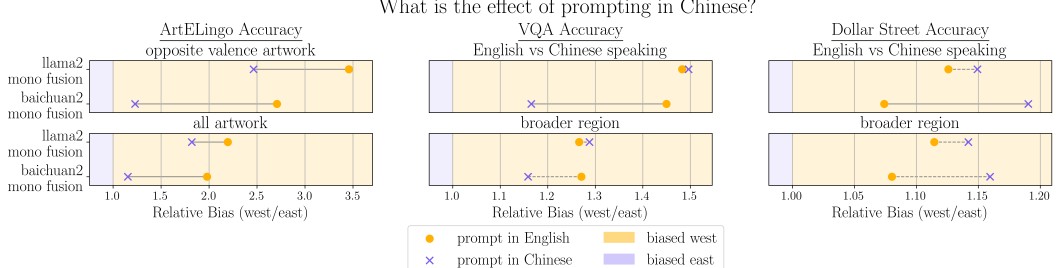

(b) ● indicates prompting in English, ✕ in Chinese. **While prompting in Chinese reduces Western bias, the reductions are larger when Chinese was well-represented during text-only pre-training.**

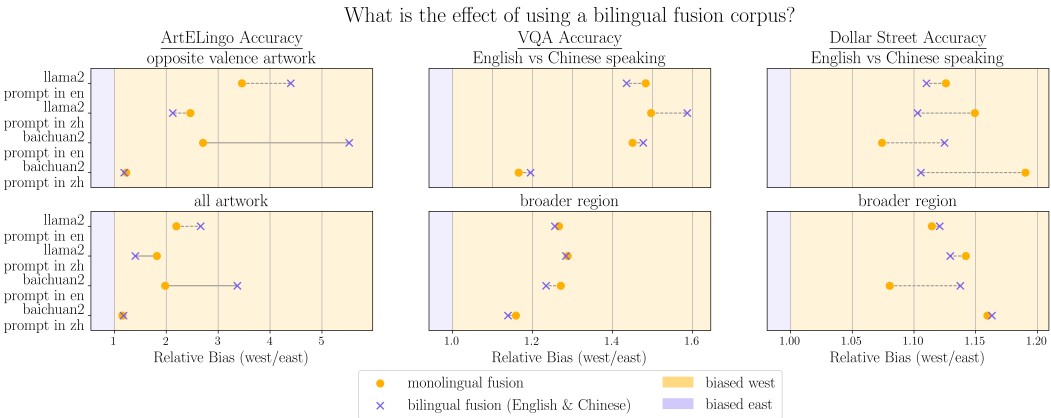

(c) ● indicates a monolingual fusion corpus, ✕ bilingual (English and Chinese). **On subjective tasks, a bilingual fusion corpus ties a prompt language to the visual perspective of its speakers, especially for English.**

Figure 5: The change in bias (Western performance divided by East Asian performance) of each our **mLLaVA** variants. Markers that fall in the gold region indicate a Western bias; in the purple region, an East Asian bias. Unbroken lines indicate bias reductions that are significant at the $\alpha = 0.05$ level.

of $1.27 \to 1.16$ on the "broader region" subset. On the subjective ArtELingo, our Llama2-based VLM sees significant bias reductions of $3.45 \to 2.47$ and $2.20 \to 1.82$ on the "opposite valence" and "all" subsets when prompting in Chinese. However, for our Baichuan2-based VLM, reductions are even larger: $2.72 \to 1.23$ and $1.98 \to 1.16$ on "opposite valence" and "all" respectively.

### 5.2.3 HOW DOES THE FUSION LANGUAGE MIX AFFECT BIAS IN IMAGE UNDERSTANDING?

Finally, our results show that **training on a balanced language mix during multimodal fusion yields no consistent change in Western bias in objective tasks**. However, on subjective tasks, a bilingual fusion corpus results **in a stronger alignment between the prompting language and the perspective of its speakers, especially for English and the Western perspective** (Figure 5c).

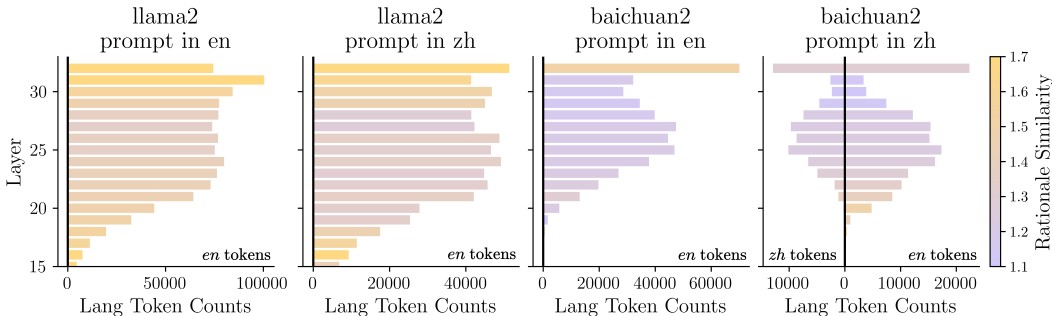

Figure 6: We map hidden states from our mLLaVA VLMs for images in ArtELingo's *opp val* subset to their nearest tokens (Nostalgebraist, 2020). Despite generating fluent Chinese, our Llama2 VLM hidden states map to English; in our Baichuan2 VLM, they map to both English and Chinese. Moreover, in both Baichuan2 VLMs, mapped tokens are more similar to ArtELingo's Chinese rationales.

With Llama2, including Chinese during multimodal fusion increases Western bias on ArtELingo ("all") when prompting in English (from $2.19 \rightarrow 2.65$) but including English decreases bias on the "opposite valence" and "all" subsets when prompting in Chinese ($2.47 \rightarrow 2.12$, $1.82 \rightarrow 1.40$). With the bilingual Baichuan2, including Chinese during multimodal fusion increases Western bias on both subsets of ArtELingo when prompting in English ($2.71 \rightarrow 5.50$, $1.98 \rightarrow 3.40$) but including English leaves Western bias unchanged when prompting in Chinese.

### 5.2.4 WHY DO BAICHUAN2 VARIANTS GENERATE MORE CULTURALLY ALIGNED CHINESE?

The much larger reductions in bias we see from using Chinese to prompt Baichuan2 suggests that while Llama2 generates fluent Chinese, it is not effectively modeling the visual perspective of a Chinese language speaker. In Figure 6, we explore this mechanistically by decoding hidden states from our Llama2 and Baichuan2 mLLaVA variants for images in ArtELingo's *opposite valence* subset using the logit lens (Nostalgebraist, 2020). For Llama2, hidden states only decode to English, even when trained to generate Chinese, while for Baichuan2, they decode to both English and Chinese. Moreover, the decoded hidden states from Baichuan2 are more similar to the rationales provided by Chinese language annotators for their annotations than the decoded hidden states from their Llama2 counterparts; unlike Llama2, Baichuan2 actually leverages associations that are culturally Chinese. Thus, our work provides corroborating evidence in image understanding (both *intrinsic*, via probing, and *extrinsic*, via task evaluation) for a prior study that showed that Llama2 "works" in English (Wendler et al., 2024). Technical details of this probing are in Section A.4 of the Appendix.

## 6 DISCUSSION AND CONCLUSION

Our work shows that VLMs exhibit a Western bias in image understanding that stems in part from the language mix in their text-only pre-training. While we focus on Chinese because it is resourced enough for controlled experimentation, this also lends greater weight to our results: it is a floor for Western bias. We should expect the bias relative to other less-resourced cultures to be even greater.

Prior work has shown that LLMs encode cultural knowledge from their pre-training in their weights (Kassner et al., 2021). We extend this analysis to VLMs. Including more Chinese in *language pre-training* results in a VLM with reduced Western bias in *image understanding*. On objective tasks, these VLMs leverage pre-training knowledge to make sense of diverse images. And on subjective tasks, they are better models of non-Western perspectives, even when prompted in English.

Finally, while multilingual image-text retrieval (Ramos et al., 2024) and machine translation (Zhou et al., 2021; Qiu et al., 2022; Geigle et al., 2023) can both produce multilingual VLMs, our results suggest limitations to these methods. LLMs trained on naturally occurring non-English language learn cultural associations relevant to diverse image understanding. The short texts in multimodal corpora likely contain little cultural knowledge. And if machine translation is employed, we should expect even less. The Western bias of these models cannot be fixed in the final stages of VLM development – its remediation requires consideration from the start (Santy et al., 2023).

## 7 ETHICS STATEMENT

Our work relies on the simplifying assumption that both the country where an image was taken and the language in which it was annotated are reliable indicators of a fixed, underlying culture. Though this reduction is unfortunately necessary in AI research due to the lack of datasets with rich cultural annotations, it runs the risk of reinforcing stereotypes by stripping away important nuance.

Moreover, while we draw images from a variety of representative countries when constructing our Western and East Asian task subsets, our study of the role that language plays in VLM bias is limited to English and Chinese. This is due to the unavailability of comparable LLMs for other East Asian languages. While the assumption that English is closer to Western culture than Chinese (and that Chinese is closer to East Asian culture than English) helps explain the bias reductions we see in our results, we emphasize here that neither English nor Chinese perfectly reflects Western and East Asian cultures in their entirety. Both languages are widely spoken in countries around the world and both the West and East Asia are home to millions of people that speak neither.

While we are sensitive to these dimensions of our work, we still believe that studies like ours are an important first step toward addressing cultural bias in AI by providing insights regarding its source. It is our hope that future work will build on ours and bridge these remaining gaps.

Finally, all vision-language models have the potential for harmful use including but not limited to surveillance. As our focus is on understanding the modeling decisions behind bias in existing VLMs, we have no additional concerns regarding harmful use beyond those relevant to the broader field.

## 8 REPRODUCIBILITY STATEMENT

To facilitate full reproducibility of our results, we have:

1. included comprehensive technical details in the Appendix
2. when relying on an LLM-as-a-Judge, chosen a model with publicly available, fixed weights
3. published both our code and models at `https://github.com/amith-ananthram/see-it-from-my-perspective` with thorough documentation
4. made our models available to the broader research community on HuggingFace

ACKNOWLEDGEMENTS

This research is being developed in part with funding from the Defense Advanced Research Projects Agency (DARPA) Cross-Cultural Understanding (CCU) program under Contract No HR001122C0034 and the ECOLE program under Contract No HR00112390060, the National Science Foundation and by DoD OUSD (R&E) under Cooperative Agreement PHY-2229929 (The NSF AI Institute for Artificial and Natural Intelligence) and DRL-2112635 (the NSF AI Engage Institute), the Columbia Center for Artificial Intelligence and Technology (CAIT) and ONR Grant N00014-23-1-2356. The views, opinions and/or findings expressed are those of the authors and should not be interpreted as representing the official views or policies of the Department of Defense, the National Science Foundation or the U.S. Government.

Additionally, we'd like to thank Fei-Tzin Lee and our ICLR reviewers for their thoughtful feedback on earlier versions of this work.

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

# A APPENDIX

## A.1 TASK DETAILS

### A.1.1 OBJECT IDENTIFICATION

1. *English*: What object is in this image?
2. *Chinese*: 这张图片中是什么物体?

These two prompts, in English and Chinese, were hand-written by NLP graduate students who are native English and Chinese speakers. Additionally, Dollar Street's class set of 289 labels was manually translated into Chinese by a native Chinese NLP graduate student.

### A.1.2 VISUAL QUESTION ANSWERING

For visual question answering, questions and candidate answers from A-OKVQA and CVQA were concatenated together and used as prompts, for example:

1. *English*: What is the dog trying to catch? Choose exactly one from person, frisbee, kite, ball.
2. *Chinese*: 狗想抓什么? 从人、飞盘、风筝、球中选择一个。

English prompts were drawn directly from both corpora. Chinese prompts were produced from the English prompts through the Google Translate API. We asked a native Chinese NLP graduate student to rate whether or not each of random sample of 50 of these machine translations were "a good translation of the English" on a 4-point Likert scale (with 1 indicating strongly disagree, 2 indicating disagree, 3 indicating agree and and 4 indicating strongly agree). They received an average rating of 3.82 indicating high quality.

### A.1.3 ART EMOTION CLASSIFICATION

1. *English*: What emotion does this work of art evoke?

2. *Chinese*: 这件艺术作品唤起了什么样的情感?

These two prompts, in English and Chinese, were hand-written by NLP graduate students who are native English and Chinese speakers.

### A.2 TASK EVALUATION DETAILS

### A.2.1 OBJECT IDENTIFICATION

We judge our English and Chinese language generations for our Object Identification task against their correct labels in a text-only setting using `Prometheus-2-8x7B` (Kim et al., 2024). We use the following rubric in Prometheus to produce numerical scores for each generation.

```
instruction = Identify the object(s) in this image.

criterion = Does the model's answer describe the same object (or
    kind of object) as the reference?

score1_description = The model's answer describes an object that
    is completely unrelated to the reference answer.
score2_description = The model's answer describes an object that
    is very different from the reference answer.
score3_description = The model's answer describes an object
    similar to the reference answer.
score4_description = The model's answer describes a specific
    instance of the reference answer.
score5_description = The model's answer describes the same object
    as the reference answer
```

To improve the quality of these judgments, we apply self-consistency (Wang et al., 2022): each model generation is judged 5 times (with a temperature of 1.0 and a nucleus sampling threshold of 0.9). We consider the mean score and assign a model credit if it is greater than or equal to 3.0.

We validate the effectiveness of `Prometheus-2-8x7B` as judge for object identification through a comparative human evaluation. We ask two native English speakers and two native Chinese speakers, all of whom are graduate students in NLP, to manually judge a random sample 100 English and 100 Chinese model generations respectively. Specifically, we ask them to assign a generation credit if "the generation describes an object listed in the label". They complete these annotations without access to the source image and consider a concatenation of all correct labels for each image.

In the Table 2, we present human-human and human-machine agreement numbers in English and Chinese across representative samples from our Western (the United States) and East Asian (China) splits. For English generations, we find that Prometheus exceeds human-human agreement with a mean agreement of 89% and a mean Cohen's $\kappa$ of 0.65 when compared to each human annotator individually. We observe similar mean agreement / mean Cohen's $\kappa$ for Chinese generations though in this case these values are lower than the human-human setting (95% and 0.85).

Table 2: Human-human and mean human-machine agreement (percent agreement and Cohen's $\kappa$) judging 200 open-ended generations on the Object Identification task where images are sampled from the United States and China subsets of Dollar Street (Rojas et al., 2022).

| Prompt Language | Image Country | Human | | Prometheus-2 8x7B | |
|---|---|---|---|---|---|
| | | % | $\kappa$ | % | $\kappa$ |
| English | | 0.83 | 0.51 | 0.89 | 0.65 |
| English | United States | 0.82 | 0.36 | 0.91 | 0.65 |
| English | China | 0.84 | 0.61 | 0.86 | 0.63 |
| Chinese | | 0.95 | 0.85 | 0.88 | 0.66 |
| Chinese | United States | 0.92 | 0.77 | 0.82 | 0.55 |
| Chinese | China | 0.98 | 0.94 | 0.92 | 0.80 |

### A.2.2  VISUAL QUESTION ANSWERING

We first translate all Chinese generations into English using the Google Translate API. Then, for each question-generation pair, we use `Mistral-7B-Instruct-v0.2` (Jiang et al., 2023) to extract the nearest candidate answer using the following prompt:

```
"Question: [QUESTION], Answer: [GENERATION in ENGLISH]" Which of
the following answers is closest to the one in the previous text?
1) [CANDIDATE\_1 in ENGLISH] 2) [CANDIDATE\_2 in ENGLISH]
3) [CANDIDATE\_3 in ENGLISH] or 4) [CANDIDATE\_4 in ENGLISH].
Respond concisely with a single number.
```

If a model's response contains multiple answers, we choose the first that occurs in the generation. If it contains no answers, a candidate is chosen at random.

### A.2.3  ART EMOTION CLASSIFICATION

For each model generation in English and in Chinese, we use `Mistral-7B-Instruct-v0.2` (Jiang et al., 2023) to extract the most similar emotion using the following prompt:

```
"[GENERATION]" Which of the following emotions is closest to
the one described in the previous text: amusement, anger,
awe, contentment, disgust, excitement, fear, sadness. Answer
in a single word.
```

If a model's response contains multiple answers, we choose the first that occurs in the generation. If it contains no answers, a candidate is chosen at random.

### A.2.4  MODEL REFUSALS

For our mLLaVA variants, we measured the fraction of generations that were refusals. To identify such cases, we counted all generations that were 1) an empty string or 2) judged to be a refusal by another LLM, `Mistral-7B-Instruct-v0.3`, using the following prompt:

```
"[GENERATION]" Does the quoted text contain a refusal to answer?
Please respond in a single word, \"yes\" or \"no\".
```

We found that less than 0.1% of our generations met either of these refusal criteria.

### A.3  mLLaVA TRAINING

### A.3.1  LLAMA2 VS BAICHUAN2

In training our mLLAVA variants, we strove to select base LLMs that were as close to each other as possible except along our axis of inquiry, the language mix of their pre-training corpora. This led

us to Llama2 and Baichuan2, 7 billion parameter LLMs pre-trained on around 2T tokens and then subsequently alignment tuned. While their most significant difference is the language mix of their pre-training corpora (90% English, 0.13% Chinese for Llama2 and bilingual English & Chinese for Baichuan2), in Table 3, we compare their architectural features.

Table 3: Architectural features of Llama2 and Baichuan2.

| Feature | Llama2 | Baichuan2 |
|---|---|---|
| BPE? | yes | yes |
| Split Digits? | yes | yes |
| Context Length | 4K | 4K |
| Vocabulary Size | 32K | 125K |
| Positional Embeddings | rotary | rotary |
| Activation | SwiGLU | SwiGLU |
| Layers | 32 | 32 |
| Attention Heads | 32 | 32 |
| Precision | float16 | bfloat16 |

Architecturally, the two models are very close. Their most significant difference is the vocabulary size of their tokenizers. These tokenizers were learned to accomodate each model's pre-training corpus (mostly English for Llama2 and bilingual English / Chinese for Baichuan2). While we know that the size of these pre-training corpora were similar, neither technical report provides detail on their composition beyond language mix. Moreover, both models were alignment tuned through supervised fine-tuning (SFT) and reinforcement learning from human feedback (RLHF). While the SFT corpora for both models are similarly sized (between 50k and 100k), the Baichuan2 paper does not provide details on their preference tuning beyond saying that their reward model "has a performance consistent with that of Llama2" (Yang et al., 2023). Given what has been published about both models (and other English and Chinese LLMs), we felt that Llama2 and Baichuan2 were the most comparable open weight LLMs for investigating the role of language in image understanding.

### A.3.2 CHINESE FUSION CORPUS

We use the pre-training and visual instruction tuning corpora from the original LLaVA paper as our English multimodal fusion corpus. For Chinese, we use a publicly available machine translation of this corpus from: `https://huggingface.co/datasets/LinkSoul/Chinese-LLaVA-Vision-Instructions`.

To validate the quality of this machine translation, we asked an NLP graduate student and native Chinese speaker to rate whether or not each of random sample of 25 pre-training examples and 25 visual instruction tuning examples were "a good translation of the English" on a 4-point Likert scale (with 1 indicating strongly disagree, 2 indicating disagree, 3 indicating agree and and 4 indicating strongly agree). They received an average rating of 3.76 indicating high quality.

### A.3.3 TRAINING DETAILS

We use the LORA (Hu et al., 2021) training recipe published in the LLaVA repository to train each **mLLaVA** variant in 2 phases: multimodal pre-training and visual instruction tuning. This allows us to freeze our base LLMs, mitigating catastrophic forgetting of pre-trained knowledge (Kirkpatrick et al., 2017) and enabling training each variant on 4 NVIDIA A100 GPUs in a single day.

### A.4 mLLaVA PROBING

### A.4.1 THE LOGIT LENS

The logit lens is a tool from mechanistic interpretability used to probe LMs (Nostalgebraist, 2020). It works by applying the LM's "unembedding matrix" directly to hidden states from each layer of the model, projecting each hidden state to a distribution over the model's output vocabulary. By decoding an LM's hidden states in this manner, we get intrinsic insight into how a model converges

to its final output distribution through iterative updates to token probabilities. Wendler et al. (2024) make us of this technique in probing the internals of Llama2 when generating non-English text to make the case that the model is "operating" in English. Here, we perform a similar analysis of our Baichuan2 and Llama2-based VLMs to probe how each VLM handles the same input images.

### A.4.2  PROBING DETAILS

To create Figure 6, for a particular **mLLaVA** variant, we embed every image in ArtELingo's *opposite valence* subset, extracting hidden states from each layer of its base LLM. Then, we decode these hidden states using the logit lens (Nostalgebraist, 2020); we consider only the tokens with decoded probability greater than $40\%$. This results in a set of subword tokens for each image & layer. We map these tokens to the most frequent word on Wikipedia that contains that token[3], producing a list of words (in English and Chinese) for each layer. For a given layer, we count the number of English and Chinese words across all images to produce the "Lang Token Counts" in Figure 6.

Then, for each image, we use XLM-R (Conneau et al., 2019) to compute similarities between each of its decoded words at a particular layer and its English and Chinese language annotation rationales (included in ArtELingo). We take the mean of its similarities with English rationales and the mean of its similarities with Chinese rationales to produce per-layer, per-language similarity scores for each image. We average the per-language scores across all images for every layer, computing the ratio between the two. This value is depicted by the "Rationale Similarity" coloring in Figure 6.

We include the raw English and Chinese token counts and rationale similarity ratios at each layer in Table 4. A similarity greater than $1$ indicates that the tokens at that layer were more similar to the English rationales (and less than 1, more similar to the Chinese rationales).

---

[3]Using         frequency         lists         from         `https://github.com/IlyaSemenov/`
`wikipedia-word-frequency`

Table 4: Probing results. For each layer and model, "en" indicates the number of decoded English tokens, "zh" indicates the number of decoded Chinese tokens and "sim" indicates the average token similarity with English rationales divided by the average token similarity with Chinese rationales.

| Layer | llama2 (en) | | | llama2 (zh) | | | baichuan2 (en) | | | baichuan2 (zh) | | |
|---|---|---|---|---|---|---|---|---|---|---|---|---|
| | en | zh | sim | en | zh | sim | en | zh | sim | en | zh | sim |
| 1 | 72 | 0 | 2.45 | 52 | 0 | 1.34 | 0 | 0 | — | 0 | 0 | — |
| 2 | 69 | 0 | 2.65 | 107 | 0 | 1.61 | 0 | 0 | — | 0 | 0 | — |
| 3 | 74 | 0 | 2.87 | 59 | 0 | 1.54 | 0 | 0 | — | 0 | 0 | — |
| 4 | 51 | 0 | 3.55 | 66 | 0 | 1.33 | 0 | 0 | — | 0 | 0 | — |
| 5 | 83 | 0 | 1.87 | 59 | 0 | 1.27 | 0 | 0 | — | 0 | 0 | — |
| 6 | 148 | 0 | 1.67 | 98 | 0 | 1.28 | 0 | 0 | — | 0 | 0 | — |
| 7 | 162 | 0 | 1.23 | 157 | 0 | 1.39 | 0 | 0 | — | 0 | 0 | — |
| 8 | 298 | 0 | 0.96 | 308 | 0 | 1.36 | 0 | 0 | — | 0 | 0 | — |
| 9 | 364 | 0 | 1.11 | 2671 | 0 | 1.44 | 0 | 0 | — | 0 | 0 | — |
| 10 | 542 | 0 | 1.21 | 3743 | 0 | 1.24 | 0 | 0 | — | 0 | 0 | — |
| 11 | 704 | 0 | 1.15 | 4387 | 0 | 1.34 | 1 | 0 | 5.16 | 0 | 0 | — |
| 12 | 982 | 0 | 1.27 | 3969 | 0 | 1.27 | 4 | 0 | 1.67 | 0 | 0 | — |
| 13 | 1645 | 0 | 2.50 | 3906 | 0 | 1.22 | 15 | 0 | 2.58 | 0 | 0 | — |
| 14 | 2352 | 0 | 1.16 | 3536 | 0 | 1.15 | 7 | 0 | 2.26 | 7 | 0 | 0.88 |
| 15 | 2472 | 0 | 1.31 | 5721 | 0 | 1.50 | 6 | 0 | 1.09 | 3 | 0 | 0.89 |
| 16 | 4613 | 0 | 1.72 | 6798 | 0 | 1.43 | 111 | 0 | 1.68 | 33 | 0 | 1.55 |
| 17 | 7541 | 0 | 1.93 | 9277 | 0 | 1.78 | 136 | 0 | 1.67 | 68 | 0 | 1.89 |
| 18 | 11298 | 0 | 1.58 | 11424 | 0 | 1.55 | 382 | 0 | 1.35 | 218 | 1 | 2.02 |
| 19 | 19579 | 0 | 1.55 | 17552 | 0 | 1.46 | 590 | 0 | 1.42 | 361 | 2 | 1.61 |
| 20 | 32499 | 0 | 1.51 | 25341 | 0 | 1.34 | 1641 | 21 | 1.10 | 983 | 14 | 1.40 |
| 21 | 44263 | 11 | 1.49 | 27854 | 5 | 1.33 | 5782 | 120 | 1.23 | 4811 | 237 | 1.50 |
| 22 | 64210 | 7 | 1.44 | 42098 | 0 | 1.37 | 13039 | 80 | 1.29 | 8514 | 1155 | 1.39 |
| 23 | 73047 | 10 | 1.44 | 45719 | 1 | 1.32 | 19733 | 50 | 1.23 | 10185 | 1833 | 1.34 |
| 24 | 76269 | 1 | 1.42 | 44809 | 0 | 1.30 | 26892 | 19 | 1.22 | 11361 | 4925 | 1.29 |
| 25 | 79960 | 0 | 1.41 | 49209 | 0 | 1.35 | 37796 | 5 | 1.21 | 16173 | 6569 | 1.26 |
| 26 | 75188 | 0 | 1.36 | 46590 | 0 | 1.32 | 46871 | 1 | 1.21 | 17342 | 10140 | 1.23 |
| 27 | 76809 | 1 | 1.39 | 48820 | 3 | 1.36 | 44590 | 0 | 1.20 | 15195 | 8659 | 1.23 |
| 28 | 73872 | 0 | 1.35 | 42260 | 4 | 1.27 | 47456 | 0 | 1.20 | 15377 | 9684 | 1.22 |
| 29 | 77020 | 0 | 1.39 | 41425 | 3 | 1.28 | 39814 | 0 | 1.16 | 12220 | 7421 | 1.22 |
| 30 | 77346 | 0 | 1.48 | 45040 | 3 | 1.46 | 34484 | 0 | 1.14 | 7454 | 4578 | 1.13 |
| 31 | 84294 | 1 | 1.57 | 46830 | 0 | 1.51 | 28616 | 0 | 1.16 | 3838 | 2328 | 1.11 |
| 32 | 100456 | 0 | 1.80 | 41390 | 1 | 1.62 | 32156 | 0 | 1.18 | 3378 | 2591 | 1.13 |
| 33 | 74435 | 0 | 1.78 | 51387 | 135 | 1.76 | 70039 | 0 | 1.51 | 22363 | 12905 | 1.30 |

Table 5: The scores, biases and bias reductions of our off-the-shelf (OTS) VLMs on the Western ($W$) and East Asian ($E$) splits of our selected Dollar Street (DS), VQA and ArtELingo (Art) subsets. Note that as blip2-flan-t5-xxl does not generate fluent Chinese, we do not evaluate it in that setting.

| | Model | Prompt in en | | | Prompt in zh | | | Bias Reduction |
|---|---|---|---|---|---|---|---|---|
| | | $W$ | $E$ | Bias | $W$ | $E$ | Bias | |
| DS: en/zh | blip2-flan-t5-xxl | 0.716 | 0.654 | 1.094 | — | — | — | — |
| | llava-1.5-7b | 0.728 | 0.612 | 1.189 | 0.541 | 0.492 | 1.101 | 0.088 |
| | llava-1.5-mistral-7b | 0.824 | 0.729 | 1.130 | 0.007 | 0.006 | 1.156 | −0.026 |
| | mblip-bloomz-7b | 0.638 | 0.578 | 1.103 | 0.513 | 0.479 | 1.071 | 0.032 |
| | qwen-vl-7b | 0.700 | 0.608 | 1.151 | 0.516 | 0.541 | 0.953 | 0.198 |
| | qwen-vl-chat-7b | 0.700 | 0.624 | 1.122 | 0.616 | 0.533 | 1.155 | −0.034 |
| DS: region | blip2-flan-t5-xxl | 0.681 | 0.620 | 1.097 | — | — | — | — |
| | llava-1.5-7b | 0.692 | 0.602 | 1.149 | 0.528 | 0.470 | 1.123 | 0.026 |
| | llava-1.5-mistral-7b | 0.803 | 0.701 | 1.147 | 0.004 | 0.004 | 0.897 | 0.250 |
| | mblip-bloomz-7b | 0.615 | 0.551 | 1.116 | 0.494 | 0.451 | 1.095 | 0.021 |
| | qwen-vl-7b | 0.666 | 0.577 | 1.154 | 0.494 | 0.497 | 0.993 | 0.162 |
| | qwen-vl-chat-7b | 0.656 | 0.591 | 1.111 | 0.605 | 0.499 | 1.214 | −0.104 |
| VQA: en/zh | blip2-flan-t5-xxl | 0.686 | 0.553 | 1.240 | — | — | — | — |
| | llava-1.5-7b | 0.371 | 0.367 | 1.013 | 0.507 | 0.383 | 1.324 | −0.311 |
| | llava-1.5-mistral-7b | 0.491 | 0.463 | 1.060 | 0.521 | 0.392 | 1.329 | −0.269 |
| | mblip-bloomz-7b | 0.605 | 0.402 | 1.506 | 0.536 | 0.424 | 1.263 | 0.242 |
| | qwen-vl-7b | 0.762 | 0.621 | 1.227 | 0.632 | 0.592 | 1.069 | 0.158 |
| | qwen-vl-chat-7b | 0.774 | 0.611 | 1.267 | 0.631 | 0.566 | 1.116 | 0.151 |
| VQA: region | blip2-flan-t5-xxl | 0.580 | 0.488 | 1.187 | — | — | — | — |
| | llava-1.5-7b | 0.361 | 0.337 | 1.073 | 0.412 | 0.312 | 1.321 | −0.248 |
| | llava-1.5-mistral-7b | 0.448 | 0.410 | 1.092 | 0.436 | 0.356 | 1.226 | −0.134 |
| | mblip-bloomz-7b | 0.485 | 0.346 | 1.402 | 0.450 | 0.358 | 1.260 | 0.142 |
| | qwen-vl-7b | 0.655 | 0.527 | 1.244 | 0.533 | 0.473 | 1.126 | 0.118 |
| | qwen-vl-chat-7b | 0.648 | 0.521 | 1.245 | 0.514 | 0.444 | 1.158 | 0.087 |
| Art: opp. val. | blip2-flan-t5-xxl | 0.220 | 0.111 | 1.983 | — | — | — | — |
| | llava-1.5-7b | 0.182 | 0.145 | 1.252 | 0.081 | 0.064 | 1.270 | −0.018 |
| | llava-1.5-mistral-7b | 0.193 | 0.152 | 1.272 | 0.082 | 0.073 | 1.114 | 0.158 |
| | mblip-bloomz-7b | 0.098 | 0.100 | 0.980 | 0.074 | 0.071 | 1.049 | −0.068 |
| | qwen-vl-7b | 0.179 | 0.160 | 1.118 | 0.089 | 0.079 | 1.131 | −0.013 |
| | qwen-vl-chat-7b | 0.217 | 0.152 | 1.426 | 0.083 | 0.084 | 0.998 | 0.428 |
| Art: all | blip2-flan-t5-xxl | 0.268 | 0.261 | 1.027 | — | — | — | — |
| | llava-1.5-7b | 0.269 | 0.262 | 1.025 | 0.036 | 0.033 | 1.087 | −0.062 |
| | llava-1.5-mistral-7b | 0.312 | 0.257 | 1.213 | 0.044 | 0.036 | 1.229 | −0.016 |
| | mblip-bloomz-7b | 0.160 | 0.166 | 0.962 | 0.059 | 0.035 | 1.696 | −0.734 |
| | qwen-vl-7b | 0.267 | 0.255 | 1.048 | 0.054 | 0.042 | 1.285 | −0.236 |
| | qwen-vl-chat-7b | 0.266 | 0.247 | 1.079 | 0.059 | 0.031 | 1.914 | −0.835 |

Table 6: The accuracy, Western bias and bias reductions of our mLLaVA variants. $M$ indicates the base LLM (L: the mostly English Llama2, B: the English/Chinese Baichuan2); its subscript indicates the fusion corpus (m: monolingual, bi: bilingual). $L$, the prompting language. $W$ and $E$, the accuracy on the Western and East Asian splits. Bias, the Western bias (West / East Asian). $\Delta_{llm}$, $\Delta_{lang}$ and $\Delta_{fus}$ indicate the change in Western bias when changing the base LLM to Baichuan2, the prompting language to Chinese or the fusion corpus to bilingual. $p$ lists $p$-values calculated using a two-factor design (Krzywinski, 2014). **Bolded** entries are significant at the $\alpha = 0.05$ level.

| | $M$ | $L$ | $W$ | $E$ | Bias | $\Delta_{llm}$ | $p$ | $\Delta_{lang}$ | $p$ | $\Delta_{fus}$ | $p$ |
|---|---|---|---|---|---|---|---|---|---|---|---|
| Dollar Street: en/zh | $L_m$ | en | 0.812 | 0.721 | 1.126 | | | | | | |
| | | zh | 0.608 | 0.529 | 1.149 | | | −0.023 | 0.676 | | |
| | $B_m$ | en | 0.757 | 0.705 | 1.074 | 0.052 | 0.143 | | | | |
| | | zh | 0.697 | 0.585 | 1.190 | −0.041 | 0.274 | **−0.116** | **0.035** | | |
| | $L_{bi}$ | en | 0.825 | 0.743 | 1.110 | | | | | 0.016 | 0.722 |
| | | zh | 0.594 | 0.538 | 1.103 | | | | | 0.046 | 0.433 |
| | $B_{bi}$ | en | 0.739 | 0.657 | 1.125 | | | | | −0.051 | 0.278 |
| | | zh | 0.652 | 0.590 | 1.106 | | | | | 0.085 | 0.094 |
| Dollar Street: region | $L_m$ | en | 0.782 | 0.702 | 1.114 | | | | | | |
| | | zh | 0.597 | 0.523 | 1.142 | | | −0.028 | 0.740 | | |
| | $B_m$ | en | 0.743 | 0.688 | 1.080 | 0.034 | 0.147 | | | | |
| | | zh | 0.654 | 0.564 | 1.159 | −0.017 | 0.416 | −0.079 | 0.059 | | |
| | $L_{bi}$ | en | 0.802 | 0.715 | 1.121 | | | | | −0.006 | 0.715 |
| | | zh | 0.595 | 0.527 | 1.130 | | | | | 0.012 | 0.751 |
| | $B_{bi}$ | en | 0.728 | 0.640 | 1.138 | | | | | −0.057 | 0.067 |
| | | zh | 0.646 | 0.555 | 1.163 | | | | | −0.004 | 0.975 |
| VQA: en/zh | $L_m$ | en | 0.644 | 0.434 | 1.483 | | | | | | |
| | | zh | 0.510 | 0.341 | 1.496 | | | −0.013 | 0.361 | | |
| | $B_m$ | en | 0.620 | 0.428 | 1.450 | 0.033 | 0.696 | | | | |
| | | zh | 0.555 | 0.476 | 1.165 | **0.331** | **0.044** | **0.285** | **0.011** | | |
| | $L_{bi}$ | en | 0.521 | 0.363 | 1.435 | | | | | 0.048 | 0.244 |
| | | zh | 0.485 | 0.305 | 1.587 | | | | | −0.090 | 0.822 |
| | $B_{bi}$ | en | 0.565 | 0.383 | 1.477 | | | | | −0.027 | 0.822 |
| | | zh | 0.519 | 0.434 | 1.195 | | | | | −0.030 | 0.894 |
| VQA: region | $L_m$ | en | 0.515 | 0.407 | 1.266 | | | | | | |
| | | zh | 0.412 | 0.320 | 1.288 | | | −0.022 | 0.518 | | |
| | $B_m$ | en | 0.494 | 0.389 | 1.270 | −0.005 | 0.908 | | | | |
| | | zh | 0.448 | 0.387 | 1.159 | 0.129 | 0.218 | 0.112 | 0.082 | | |
| | $L_{bi}$ | en | 0.440 | 0.350 | 1.256 | | | | | 0.010 | 0.461 |
| | | zh | 0.390 | 0.304 | 1.284 | | | | | 0.004 | 0.809 |
| | $B_{bi}$ | en | 0.472 | 0.383 | 1.234 | | | | | 0.037 | 0.532 |
| | | zh | 0.432 | 0.379 | 1.139 | | | | | 0.012 | 0.727 |
| ArtELingo: opp. val. | $L_m$ | en | 0.249 | 0.072 | 3.448 | | | | | | |
| | | zh | 0.207 | 0.084 | 2.472 | | | **0.976** | **0.000** | | |
| | $B_m$ | en | 0.241 | 0.089 | 2.715 | 0.733 | 0.081 | | | | |
| | | zh | 0.306 | 0.249 | 1.228 | **1.244** | **0.000** | **1.487** | **0.000** | | |
| | $L_{bi}$ | en | 0.220 | 0.050 | 4.389 | | | | | −0.941 | 0.625 |
| | | zh | 0.185 | 0.087 | 2.124 | | | | | 0.349 | 0.055 |
| | $B_{bi}$ | en | 0.221 | 0.040 | 5.505 | | | | | **−2.790** | **0.032** |
| | | zh | 0.288 | 0.242 | 1.190 | | | | | 0.039 | 0.526 |
| ArtELingo: all | $L_m$ | en | 0.292 | 0.133 | 2.195 | | | | | | |
| | | zh | 0.233 | 0.128 | 1.822 | | | **0.373** | **0.000** | | |
| | $B_m$ | en | 0.291 | 0.147 | 1.981 | **0.213** | **0.000** | | | | |
| | | zh | 0.367 | 0.318 | 1.156 | **0.666** | **0.000** | **0.825** | **0.000** | | |
| | $L_{bi}$ | en | 0.245 | 0.092 | 2.647 | | | | | −0.452 | 0.076 |
| | | zh | 0.240 | 0.171 | 1.401 | | | | | **0.421** | **0.000** |
| | $B_{bi}$ | en | 0.219 | 0.065 | 3.393 | | | | | **−1.411** | **0.004** |
| | | zh | 0.267 | 0.227 | 1.178 | | | | | −0.022 | 0.053 |

