# OpenReview forum: "See It from My Perspective: How Language Affects Cultural Bias in Image Understanding"
_ICLR.cc/2025/Conference — ICLR 2025 Poster_

### Official Review · Reviewer_pN9q · 2024-11-02

**Soundness:** 3
**Presentation:** 3
**Contribution:** 3
**Rating:** 6
**Confidence:** 4

**Summary:**

This paper investigates the Western bias in vision-language models (VLMs) and the role of language in cultural bias within image understanding. It evaluates VLMs on tasks with culturally diverse images and annotations, finding a Western bias that can be reduced by more diverse language representation during pre-training. The paper highlights the importance of multilingual foundation models for equitable VLMs and provides empirical evidence that a balanced language mix in pre-training significantly reduces bias.

**Strengths:**

* The paper discusses the cultural bias in VLMs, providing a comprehensive analysis of how language influences this bias in VLMs.
* The paper offers a comparative analysis between English and Chinese, two major languages, which provides valuable insights into bias reduction.
* The paper presents a comprehensive methodology, with controlled experiments to trace bias sources and measure the effects of different language modeling choices on bias.

**Weaknesses:**

* The study's focus on English and Chinese may limit the generalizability of the findings to other languages and cultures.
* While the paper provides evidence of bias reduction, the authors need to do a deeper analysis of why certain models exhibit less bias, potentially through a more detailed examination of their training data and processes.

**Questions:**

* Could the authors elaborate on how their findings on language mix during pre-training could be extended to other non-Western languages and cultures?
* The findings in the paper are somewhat conventional and reflect existing consensus in LLMs. What do the authors think the same findings reflect about the relationship between vision(image) and language(text)?

I look forward to an active discussion with the authors during the rebuttal phase and will revise my score accordingly.

**Details Of Ethics Concerns:**

The paper don't have ethical concerns.

---

> ### Author Response · Authors · 2024-11-21
> **Response to Reviewer pN9q, Part 1 / 2**
>
> We’d like to thank you for your thoughtful comments.  We especially appreciate that you found our work to be comprehensive in its analysis and valuable in its insights.
>
> We’ve included responses to your questions below along with the line numbers of relevant areas of the paper where answers can be found.  Unless otherwise specified, all line numbers refer to the updated version of our paper (where newly added content appears in **blue**).
>
> 1) **Other languages and cultures**
>
>    We agree that we would love to see our work extended to more languages and cultures.  While we were able to draw evaluation examples from both the narrow set of English/Chinese speaking countries and the broader set of their geographical Western/East Asian neighbors to establish Western/East Asian bias (see Table 1, lines 254 \- 264 of both our original and revised submissions), we are limited in our ability to explore the effects of other languages because of the lack of comparable LLMs trained on equivalent amounts of English/Chinese/other language text.  We emphasize here that comparability is key to our analysis as it allows us to make more confident claims about the effects of different language modeling choices on downstream bias in image understanding.  With comparable LLMs, we have the computational resources to do multimodal post-training but unfortunately (as an academic group) we do not have the resources to train comparable LLMs in other languages.
>
>    Still, we believe that our experiments leave us with a good sense of what we can expect if our work were extended to other languages and cultures.  As Chinese is the most resourced non-Western language online, we suspect the bias (and effects on bias) we measure in our paper should be a floor for bias in other cultures – we should expect bias in those settings to be worse.  However, we argue that we needn’t accept this outcome.  Multilingual VLMs should be built on LLMs that are trained on a large amount of naturally occurring multilingual text – in particular, model developers should leverage as much of the unimodal text as is available online in languages that are spoken by members of their target culture.
>
>
> 2) **Bias sourcing**
>
>    As described above, the challenge in isolating the effects of language modeling choices on bias in image understanding is in ensuring that the base LLMs we work with are comparable except in the language mix used in their construction.  In selecting Llama2 and Baichuan2, we strove to do just that.  We’ve added a section to the Appendix of our revised submission (section A.3.1, lines 902 \- 929\) that compares the architectures and construction of these two models beyond parameter and token counts.  You’ll see that the models are very close with the exception of the vocabulary size of their tokenizers (which were learned to accommodate each model’s specific pre-training corpus, English vs English & Chinese).  Given the similarity of these two models (in architecture and pre-training), we believe our work effectively measures the effects of language on culturally informed visual tasks.

---

> > ### Author Response · Authors · 2024-11-21
> > **Response to Reviewer pN9q, Part 2 / 2**
> >
> > 3) **Relationship between vision and language**
> >
> >     We believe that deep image understanding is difficult to solve by focusing on the image distribution alone.  Text, especially the longer, richer text that appears without accompanying images, contains valuable signal regarding how we associate the kinds of objects that appear in images.
> >
> >     In our work, we show that language-specific LLMs endow their VLMs with culturally relevant subsets of these associations – English-based VLMs are more likely to encode culturally Western associations (e.g., a woman in a white dress is a bride) while Chinese-based VLMs are more likely to encode culturally East Asian associations (e.g., a woman in a red dress is a bride).  Moreover, we observe this phenomenon in both objective tasks and subjective tasks and demonstrate when one might be able to effectively leverage those associations for different end users (i.e., when does prompting in Chinese help?).  Prior work on bias in VLMs has focused primarily on the image distribution or in augmenting VLMs with culturally relevant knowledge bases (see our Related Work, lines 135 - 142 of both our original and revised submissions).  Our work is complementary and suggests to VLM developers that they should consider multilinguality starting from LLM construction.
> >
> > We hope these answers address your questions.  Thank you for helping us make our paper stronger with your review.

---

> > > ### Comment · Reviewer_pN9q · 2024-11-21
> > > **Respond to the responses to Paper 12153**
> > >
> > > Thanks for the further explanation and clarification of your response.

---

### Official Review · Reviewer_Z8Wd · 2024-11-03

**Soundness:** 3
**Presentation:** 3
**Contribution:** 3
**Rating:** 6
**Confidence:** 4

**Summary:**

This paper investigates the effects of language modeling in multicultural image understanding. The authors focus on English and Chinese as proxies for Western and Eastern cultures, and show how Western bias (defined as the performance ratio on Western data over Eastern data) across different tasks varies as a factor of the base LLMs, language prompt and fusion corpora. Based on performance in three tasks, the authors argue for including multilingual and multicultural data early in model development, as opposed to later stages (fusion and prompts).

**Strengths:**

1. The paper proposes a systematic evaluation of language modeling towards multicultural image understanding in VLMs: the authors use two different pretrained LLMs as backbone (one English, and one English–Chinese); prompts in English and Chinese; and fusion corpora in English-only, Chinese-only and mixed.
2. The results corroborate the findings that earlier adoption of multilingual data in LLMs consistently benefits upstream VLMs.
3. A mechanistic study shows that the hidden states of Chinese tokens of a multilingual VLM initialized with an English backbone only decode to English, which the authors suggest is due to the model not effectively modeling the visual perspective of Chinese speakers.

**Weaknesses:**

1. The caption of Fig 4 says that “western bias reductions are seen across all tasks when prompting in Chinese.” However, from Fig 4, it seems that prompting in Chinese mostly exacerbates the performance of VLMs.
2. Prior work had explored the role of language in multilingual understanding and generation in VLMs, such as [1, 2]. It would be good to discuss these papers and compare your findings with theirs.
3. It would also be good to report overall performance, in addition to bias, for these models. For example, a model that performs at random will likely have a bias of 1.
4. Nit: Please make text in Figures much bigger.

---
[1] Qiu et al. Multilingual Multimodal Learning with Machine Translated Text. EMNLP 2022.

[2] Ramos et al. PAELLA: Parameter-Efficient Lightweight Language-Agnostic Captioning Model. NAACL 2024.

**Questions:**

1. I understand that using the Chinese-translated version of the LLaVA dataset allows for a potentially better comparison with English models. But, as reinforced by your results, it would be great to see how culturally-relevant data affects performance and bias. For this, you could explore using the Wukong dataset [3].

---
[3] Gu et al. Wukong: A 100 Million Large-scale Chinese Cross-modal Pre-training Benchmark. NeurIPS 2022.

---

> ### Author Response · Authors · 2024-11-21
> **Response to Reviewer Z8Wd, Part 1 / 2**
>
> We’d like to thank you for your thoughtful comments.  We especially appreciate that you found our work to be systematic in both our experimental results and our mechanistic study.
>
> We’ve included responses to your questions below along with the line numbers of relevant areas of the paper where answers can be found.  Unless otherwise specified, all line numbers refer to the updated version of our paper (where newly added content appears in **blue**).
>
> 1) **Effect of prompting in Chinese in OTS models**
>
>    The caption in Figure 4 (lines 378 \- 389 of both our original and revised submissions) says “While Western bias reductions are seen across all tasks when prompting in Chinese, they are not consistently seen in every model.”  As you say, in some cases (15/30), prompting in Chinese increases Western bias (the purple x appears to the right of the gold circle).  However, in the remaining cases (15/30), prompting in Chinese reduces Western bias (the purple x appears to the left of the gold circle).  As such, though bias reductions are seen in a few models for every task (3 for DollarStreet, 3 for VQA and 2 for ArtELingo), they are not seen consistently (lines 368 \- 371 in our original paper).  We’ve updated the caption for Figure 4 (lines 393 in our revised submission) to emphasize this.
>
>
> 2) **Prior work on multilingual VLMs**
>
>    Thanks for linking these two papers on improving multilingualism in VLMs.  Rather than multilingualism, the focus of our work is multiculturalism – when a VLM generates Chinese, is the Chinese it’s generating culturally aligned?  For example, when identifying the emotion evoked by a work of art, beyond generating fluent Chinese, does the VLM select the emotion chosen by a majority of Chinese speaking annotators?  We visualize this in Figure 1 (lines 54 \- 67 in both our original and revised submissions).
>
>
>
>    In the first paper, Qiu et al. find that machine translation is an effective way to improve the multilingual performance of VLMs.  In contrast, our work demonstrates there are limits to the gains one can expect from machine translation – it does not bridge the cultural gap as effectively as using a base LLM pre-trained on large quantities of naturally occurring text in a culturally relevant language.
>
>
>
>    In the second paper, Ramos et al. find that parameter efficient training coupled with retrieval augmentation in a target language can allow a VLM (built with a multilingual base LLM) to transfer more effectively to a target language.  In contrast, we focus on the choice of base LLM and its effect on the cultural alignment of multilingual generations.
>
>
>
>    We’ve updated our Discussion to include both of these works (lines 534 \- 537 of our revised submission).
>
>
>
> 3) **Overall Performance**
>
>    We include the overall performance of our mLLaVA variants in Table 4 of the Appendix of the original paper (lines 925 \- 971\) , Table 5 of the updated version (lines 1087 \- 1133).  In nearly all of our evaluation settings, performance is above chance (25% for VQA, 12.5% for ArtELingo).  The one exception is for the subset of ArtELingo where some models (especially the Llama based variants) perform below chance on the Chinese split of the task (but in these cases, the models are favoring the label the English split, as can be seen in the above chance performance when judged against English labels).
>
>
>
>    We’ve updated the Appendix to include the overall performance numbers from our OTS variants (Table 4, lines 1036 \- 1072 in the revised version).  All models perform above chance with the exception of Blip2-Flan-T5-XXL when prompted in Chinese as the model cannot process or generate fluent Chinese.  We’ve also updated the paper to reference this table (lines 365 \- 366 in the revised version).

---

> > ### Author Response · Authors · 2024-11-21
> > **Response to Reviewer Z8Wd, Part 2 / 2**
> >
> > 4) **Pre-training with naturally occurring Chinese image-text data**
> >
> >    Thanks for pointing us to this interesting dataset.  In this work, we wanted to explore the effects of different language modeling choices on cultural bias in image understanding in as controlled a setting as possible.  By using a Chinese machine translation of the LLaVA pre-training corpus, we can ensure that the semantic information both our Llama2 and Baichuan2 VLMs are exposed to during multimodal fusion is identical (with the only difference being the language the information is expressed in).  While it would be an interesting undertaking, selecting a similar subset of Wukong that is comparable to LLaVA would be quite difficult and the results would be hard to interpret.
> >
> >
> >
> >    We agree that training on this kind of dataset would be an exciting direction for future work (and the resultant models would be a valuable contribution to the community given their training cost).  We hope others will be able to use our fork of the LLaVA codebase to do just that.
> >
> > 5) **Text Size in Figures**
> >
> >    Thanks for this suggestion – we’ve increased the size of the text in all of our results figures 4 (lines 378 \- 389), 5 (lines 432 \- 468\) and 6 (486 \- 496\) so they’re easier to read.  You should see this reflected in the revised version.
> >
> > We hope these answers address your questions.  Thank you for helping us make our paper stronger with your review.

---

> > > ### Author Response · Authors · 2024-11-25
> > > **Discussion Period Ending Soon (11/26)**
> > >
> > > Hello!  Thanks again for taking the time to give us such a thoughtful review.  While we completely understand that it is a very busy time for everyone, we would really appreciate it if you could reply to our response to your questions and let us know if we addressed them to your satisfaction (and consider updating your score if so).  The discussion period is ending soon (Nov. 26) and we would love the chance to answer any remaining questions you might have.  Thank you!

---

> > > > ### Comment · Reviewer_Z8Wd · 2024-11-25
> > > >
> > > > I thank the authors for their responses.
> > > >
> > > > They have partially addressed my concerns. As such I will keep my recommendation leaning towards (weak) acceptance.

---

### Official Review · Reviewer_bfSC · 2024-11-04

**Soundness:** 3
**Presentation:** 3
**Contribution:** 3
**Rating:** 6
**Confidence:** 4

**Summary:**

The paper aims to investigate if vision-language models are biased due to the language mix in the pretraining data or the instruction tuning data, by drawing inspiration from cognitive science research suggesting culture could affect visual perception. The authors measure western bias, or the ratio between performances over English-specific data and Chinese-specific data.

**Strengths:**

The paper presents a very novel perspective in understanding VLMs. The experiments seem well designed and well executed. The conclusions could inspire interesting discussions. Overall I think this is a good paper.

**Weaknesses:**

Considering English to represent the entire West and Chinese to represent the entire East Asia is over-generalizing. It is perfectly ok to discuss English bias (instead of Western bias), and it would only strengthen the paper.

The authors seem a bit eager to reach identical conclusions on both subjective tests and objective tests, to the point that they did not discuss opposite trends on Dollar Street (but did show the results visually in Figure 5(b)). In Figure 5(b), Western bias on Dollar Street becomes worse when the prompt is in Chinese. This contradicts the description in Line 428: prompting in a culturally closer language reduces Western bias on objective and subjective tasks.

It is common for the same instruction tuning data or the same prompting technique to have different performances on different tasks. VLM performance is just nuanced. [1] suggests that there are several VL skills learned by VLMs that could conflict. Hence, I do not think discussing this fact would weaken the paper. Rather, ignoring this fact would cause confusion and, in the worse case, be read as twisting data to suit a particular argument.

[1] Anthony Tiong, et al. What Are We Measuring When We Evaluate Large Vision-Language Models? An Analysis of Latent Factors and Biases. NAACL 2024.

Some additional details would be beneficial:

Section A.4 should contain a more detailed description of the logit lens, so that the paper becomes more self-contained. The numerical results should be shown as tables.

The comparison between Baichuan2 and Llama2 should contain more detail. For example, what are their architectures? What was their training data mix? Such data are missing from the main text and the appendix. It is the authors' job to support the claim that these models are essentially identical except training data.

Prompting in a culturally closer language reduces Western bias on objective and subjective image understanding tasks, especially if it was common during LLM pre-training -- what does "it" refer to?

**Questions:**

What are the architectures of Baichuan2 and Llama2? What was their training data mix? What are some other differences and similarities between the two models?

---

> ### Author Response · Authors · 2024-11-21
> **Response to Reviewer bfSC, Part 1 / 2**
>
> We’d like to thank you for your thoughtful comments.  We especially appreciate that you found our work to be novel, well designed/executed and of interest to the broader community.
>
> We’ve included responses to your questions below along with the line numbers of relevant areas of the paper where answers can be found.  Unless otherwise specified, all line numbers refer to the updated version of our paper (where newly added content appears in **blue**).
>
> 1) **Western bias**
>
>    We completely agree that in exploring Western bias and East Asian bias, it is important to evaluate models on examples drawn from countries beyond those that are majority English or Chinese speaking.  As such, on both object identification and VQA, we evaluate our off-the-shelf VLMs and our mLLaVA variants on examples from the narrow set of majority English/Chinese speaking countries and a larger set of their geographical neighbors (which we refer to as “broader region”).  We feel that this extended subset expands our analysis from English/Chinese bias (where models are biased on examples from English/Chinese speaking countries) to Western/East Asian bias (where models are biased on examples from the West / East Asia).  You can find the list of countries that comprise the “broader region” in Figure 3 (original & updated, lines 162 \- 181\) and Table 1 (original & updated, lines 254 \- 264\) and you can see the results for the “broader region” subset in Figures 4 (original & updated, lines 378 \- 389\) and 5 (original & updated, lines 432 \- 469).
>
>
>
>    Our analysis is focused on understanding the effects of using a culturally closer language (in this case, English and Chinese) during pre-training, multimodal fusion and inference on this Western/East Asian bias.  While we could not expand this set of languages to a broader set because of the unavailability of comparable LLMs pre-trained on other languages, our object of study is still Western/East Asian bias as our evaluation examples are drawn from a narrow and broad set of countries from those regions (please see our Ethics Statement, lines 546-553 in both our original submission and our revised submission, where we address this).
>
>
>
> 2) **Objective Tasks and DollarStreet**
>
>    We measure bias reductions on objective tasks using two benchmarks, DollarStreet for object identification and A-OKVQA/CVQA for scene understanding.  On A-OKVQA/CVQA, prompting in Chinese yields significant reductions in bias (from 1.45x to 1.17x) in our Baichuan2-based VLM which saw large quantities of Chinese during pre-training (Figure 5b, lines 444 \- 452 in both our original and revised submissions).  Thus, on both objective and subjective tasks, prompting in a culturally closer language does in fact reduce Western bias, especially if it was common during LLM pre-training.
>
>
>
>    On DollarStreet, we see (and show, Figure 5b, lines 444 \- 452 in both our original and revised submissions) an increase in Western bias from 1.07x to 1.19x.  While we include this result in our paper for the sake of completeness, we do not emphasize it in our conclusions as the effect size is much smaller than the reductions in bias we see on the objective VQA and the subjective ArtELingo (a bias increase of 0.11x compared to bias reductions of 0.285x and 1.487x respectively).
>
> 3) **Factors in VLM Performance**
>
>
>    Thanks for pointing us to this interesting work.  We agree that including this result can only strengthen our conclusions – namely that different VLMs have different task-specific strengths that are a function of the genres of data they see during pre-training.  Within a given task, bias manifests as disparity on the Western / East Asian splits.  Thus, rather than drawing conclusions on Western bias by comparing absolute scores or absolute score differences, we measure relative bias and relative bias differences.  We’ve expanded our discussion in lines 239-244 in our revised submission to include this reference.

---

> > ### Author Response · Authors · 2024-11-21
> > **Response to Reviewer bfSC, Part 2 / 2**
> >
> > 4) **Logit Lens**
> >
> >    In our revised submission, we’ve updated section 4 of the Appendix (lines 953 \- 960\) to include more detail on the logit lens and added Table 4 (lines 978 \- 1017\) containing the values depicted in the original probing Figure 6 (lines 486 \- 495 of the both the original and revised submissions).
> >
> > 5) **Llama2 vs Baichuan2**
> >
> >    Architecturally, the two models are very similar.  Their most significant difference is the vocabulary size of their tokenizers (32K for Llama2 and 125K for Baichuan2). These tokenizers were learned to accommodate each model’s pre-training corpus (mostly English for Llama2 and bilingual English/Chinese for Baichuan2). While we know that the size of these pre-training corpora were similar, neither technical report provides detail on their composition beyond language mix.
> >
> >
> >
> >    Moreover, both models were alignment tuned through supervised fine-tuning (SFT) and reinforcement learning from human feedback (RLHF). While the SFT corpora for both models are similarly sized (between 50k and 100k), the Baichuan2 paper does not provide details on their preference tuning beyond saying that their reward model “has a performance consistent with that of Llama2” (Yang et al., 2023).
> >
> >
> >
> >    Given what has been published about both models (and other English and Chinese LLMs), we felt that Llama2 and Baichuan2 were the most comparable open weight LLMs for investigating the role of language in image understanding.
> >
> >
> >
> >    In section 3.1 of our revised submission’s Appendix (lines 901 \- 930), we’ve added a table that lists each model’s architectural features explicitly and a treatment of their differences for other readers with this question.
> >
> > 6) **Rewordings**
> >
> >
> >    We’ve updated the wording of this section header to improve its clarity.  Please see lines 427-429 in the revised version of the paper.
> >
> > We hope these answers address your questions.  Thank you for helping us make our paper stronger with your review.

---

> > > ### Comment · Reviewer_bfSC · 2024-11-22
> > >
> > > Thank you for the detailed response, which addressed most of my questions. I'm not entirely convinced that using Chinese and images from East Asia is sufficient to represent East Asia, as the authors argued. Fixing this issue requires narrowing down the claim, which is easily doable.
> > >
> > > But I believe the overall merit of the paper still outweighs its flaws.
> > >
> > > Hence, I will maintain my score as marginally above the acceptance threshold. Given other reviews, I do not see a need to change the score.

---

### Official Review · Reviewer_WwFh · 2024-11-04

**Soundness:** 3
**Presentation:** 3
**Contribution:** 3
**Rating:** 6
**Confidence:** 4

**Summary:**

The authors present an interesting study that looks into the problem of Western bias in VLMs model and what role language plays in the disparity between model performance on western-centric datasets vs others. They study this problem by conducting controlled experiments for different visual understanding tasks (e.g., object identification, question answering, and art emotion classification). In this study they focus on the disparity in performance between English and Chinese and use a specific bias ratio to determine such error disparity. The main takeaway message from the study is that the underlying language model plays a big role and that it's important to make sure that the pretraining stage of the language model has a suitable level of language diversity to facilitate transfer to different cultures and languages.

**Strengths:**

The paper has some important strengths that I list below:

1. Very relevant topic considering that many SOTA VLMs used by the community are typically trained with English-only datasets and we don't know to what extent their pretraining can facilitate transfer to other cultures

2. Two important results provided by this paper: 1) a study of the bias ratio of open-source models out of the box; 2) a controlled study that investigates the capabilities of models after pretraining depending on their language model training.

**Weaknesses:**

Despite the strengths highlighted, the current paper falls short in certain aspects that I highlight below:

1. I find the structure of the paper a bit confusing because it uses the terms "bias sourcing" and "bias characterisation" as titles which haven't been described before. I believe that it would be more informative to use as titles the actual names of the experiments to make sure that the reader is aware of the corresponding content that will be described in that section.

2. The measure of bias that is reported in the current paper is misleading because it implicitly assumes that the reference metric is an accuracy. However, the authors do not acknowledge this in any way. Considering that novel models are purely generative, I believe it's important to clarify how this study can generalise to more general tasks that involve free-flow generation.

3. I think it's important to consider what is the effect of the model generations to the overall performance. I think the authors should state, together with the accuracy score, how many times the model actually provides a meaningful answer to the prompt rather than say something like "I cannot answer to this". I'm mentioning this mostly because when testing models trained in English with Chinese prompts, it might be the case that models might not follow the prompt at all. It would be useful to report this explicitly in your results (how many times does the model reply following the format provided?).

4. Using newer LLMs for their study would have been useful such as Llama-3 or Qwen instead of Llama-2. This is mostly because they are generally stronger LLMs compared to Llama-2

**Questions:**

N/a

---

> ### Author Response · Authors · 2024-11-21
> **Response to Reviewer WwFh, Part 1 / 2**
>
> We’d like to thank you for your thoughtful comments.  We especially appreciate that you found the work interesting / relevant and that you think our controlled investigation of the effects of language modeling choices fills a consequential gap in the literature.
>
> We’ve included responses to your questions below along with the line numbers of relevant areas of the paper where answers can be found.  Unless otherwise specified, all line numbers refer to the updated version of our paper (where newly added content appears in **blue**).
>
> 1) **“Bias Characterization” and “Bias Sourcing”**
>
>
> We’ve expanded the description of the methods in lines 154 \- 158 to explicitly introduce *Bias Characterization* and *Bias Sourcing*.  Additionally, in subsequent subheadings (3.1, 3.2, 4.2, 4.3, 5.2) where we use *Bias Characterization* and *Bias Sourcing*, we have additionally specified “Step 1” and “Step 2” respectively to more clearly link the two terms to our Approach figure (Figure 2, lines 108 \- 119).
>
> 2) **Classification vs Open-Generation Evaluation**
>
>
> We evaluate against three classification tasks (object identification, multiple choice VQA and emotion identification) that we believe are reasonable proxies for VLM behavior in open-ended generation settings for several reasons:
>
> - There is broad alignment around using such classification tasks for benchmarking model performance in the vision-language community.  For example, in the classification-focused MMLU benchmark (Hendrycks et al., 2021), the authors say “Some researchers have suggested that the future of NLP evaluation should focus on Natural Language Generation (NLG) (Zellers et al., 2020), an idea that reaches back to the Turing Test (Turing, 1950). However, NLG is notoriously difficult to evaluate and lacks a standard metric (Sai et al., 2020). Consequently, we instead create a simple-to-evaluate test that measures classification accuracy on multiple choice questions.”
> - Moreover, we believe our chosen tasks test the cultural alignment of vision-language abilities (object identification, scene understanding through VQA, affective analysis through emotion detection) that are core to generating culturally aligned long-form text.   In “What Are We Measuring When We Evaluate Large Vision-Language Models?” (Meng Huat Tiong et al., 2024), the authors show that fine-tuning on VQA produces the best transfer to other target vision-language tasks including open-ended generation.  This suggests that our evaluation (which includes VQA) is a reasonable proxy for the cultural alignment of general image-conditioned language generation.
> - Finally, we evaluate our VLMs on object identification and emotion classification in an open-ended generation setting, using an LLM-as-a-Judge to ensure correctness for the former and using an LLM to extract the nearest emotion for the latter.  As such, we believe our results can be extended to other open-ended generation tasks.
>
> While prior work has tackled multilingual captioning benchmarks like XM3600 (which we discuss in lines 156-157 of both our original and revised submissions), cultural differences in image descriptions tend to be subtle (different language speakers emphasize different image elements) and are difficult to differentiate via existing automatic captioning metrics, complicating evaluation.  This further motivated our decision to use easy-to-evaluate classification-style benchmarks.

---

> > ### Author Response · Authors · 2024-11-21
> > **Response to Reviewer WwFh, Part 2 / 2**
> >
> > 3) **Model Refusals**
> >
> >    Model refusals were extremely rare.  To quantify this, we conduct an additional experiment where we use another LLM (Mistral-7B-Instruct-v0.3) to evaluate whether or not each generation included a refusal.  We found that such refusals occurred less than 0.1% of the time across our mLLaVA variants.  We’ve added the details of this check in Section A.2.4 of our Appendix (lines 891 \- 897).
> >
> > 4) **Use of Newer LLMs**
> >
> >    Our focus in this paper is in trying to understand the effect of language modeling choices on downstream VLM performance.  As such, we strove to select base LLMs that were as similar to each other as possible except in the dimension of the language mix they saw during text-only pre-training (mostly English vs bilingual English and Chinese).  Llama2 and Baichuan2 were good candidates as they have the same number of parameters (7B) and were trained on a comparable number of tokens (\~2T) while differing from each other in the language mix of those tokens.  Newer LLMs diverge from each other in many more dimensions which makes drawing causal conclusions from their inclusion difficult.  For example, LLAMA3 does not have a 7B parameter variant (the closest is 8B parameters) and was pre-trained on 15T tokens.  While Qwen2 does have a 7B parameter variant, it was pre-trained on 7T tokens.  As such, they are not directly comparable to each other or Llama2 / Baichuan2.
> >
> >
> >
> >    We certainly agree that pre-training LLaVA variants that use Llama3 and Qwen2 as their base LLMs would be a valuable contribution to the broader community given the expense of their training on an academic compute budget.  Because the goal of our paper is in understanding and disentangling sources of bias, we chose models that are maximally similar, but by open-sourcing our fork of LLaVA we facilitate the training of additional models.  It is our hope that others will be able to build on our work to train and share additional mLLaVA variants.
> >
> >
> >
> > We hope these answers address your questions.  Thank you for helping us make our paper stronger with your review.

---

> > > ### Author Response · Authors · 2024-11-25
> > > **Discussion Period Ending Soon (11/26)**
> > >
> > > Hello!  Thanks again for taking the time to give us such a thoughtful review.  While we completely understand that it is a very busy time for everyone, we would really appreciate it if you could reply to our response to your questions and let us know if we addressed them to your satisfaction (and consider updating your score if so).  The discussion period is ending soon (Nov. 26) and we would love the chance to answer any remaining questions you might have.  Thank you!

---

> > > > ### Author Response · Authors · 2024-11-30
> > > > **Extended Discussion Period Ending Soon (12/2)**
> > > >
> > > > Hello!  Apologies for the double ping.  We thank you for the time you took to give us feedback on our submission and think the resulting changes have made our paper stronger.  If you feel we've addressed your questions to your satisfaction, we'd really appreciate it if you'd consider revisiting your score. Otherwise, we'd be happy to answer anything else in the time remaining. Thanks!

---

### Meta-Review · Area_Chair_UUhq · 2024-12-20

**Metareview:**

This paper extends the analysis of cultural bias, which has primarily been explored in the context of large language models (LLMs), to vision-language models (VLMs). A strength of the study lies in its approach to not only analyze pretrained models but also conduct deeper investigations by training models under various configurations. The insights derived from this work could enhance our understanding of cultural bias in VLMs and help alleviate its impact in the future. Recognizing this strength, all reviewers assigned a score of 6, indicating “marginally above the acceptance threshold.” In line with their evaluations, I also suggest accepting the paper.

However, as pointed out by reviewers bfSC and pN9q, the paper tends to overgeneralize biases observed in English-speaking and Chinese-speaking cultures as representative of "Western" and "Eastern" biases, respectively. Reviewer bfSC specifically stated, “Fixing this issue requires narrowing down the claim, which is easily doable.” The authors had sufficient opportunity to address this concern and revise their manuscript during the discussion period but did not do so adequately.

While ICLR is primarily a machine learning conference, this does not justify the lack of rigor in the sociocultural framing and analysis, which is a critical aspect of this study. Given the significance of this issue, I also strongly recommend that the authors revise the current manuscript to address this concern.

**Additional Comments On Reviewer Discussion:**

The authors actively participated in the discussion period, making efforts to address the reviewers' concerns. However, several issues, such as the overgeneralization problem, were not sufficiently resolved. Nonetheless, the reviewers placed slightly more weight on the strengths of the paper, which became the basis for recommending its acceptance.

---

### Decision · Program_Chairs · 2025-01-22

Accept (Poster)